# A Dual-Encoder Contrastive Learning Model for Knowledge Tracing

**DOI:** 10.3390/e27070685

**Published:** 2025-06-26

**Authors:** Yanhong Bai, Xingjiao Wu, Tingjiang Wei, Liang He

**Affiliations:** 1Laboratory of AI for Education, East China Normal University, Shanghai 200062, China; lucky_byh369@163.com; 2School of Pharmacy, East China Normal University, Shanghai 200062, China; xjwu@pharm.ecnu.edu.cn; 3School of Computer Science and Technology, East China Normal University, Shanghai 200062, China; lhe@cs.ecnu.edu.cn

**Keywords:** knowledge tracing, contrastive learning, graph neural network, data mining, deep learning

## Abstract

Knowledge tracing (KT) models learners’ evolving knowledge states to predict future performance, serving as a fundamental component in personalized education systems. However, existing methods suffer from data sparsity challenges, resulting in inadequate representation quality for low-frequency knowledge concepts and inconsistent modeling of students’ actual knowledge states. To address this challenge, we propose Dual-Encoder Contrastive Knowledge Tracing (DECKT), a contrastive learning framework that improves knowledge state representation under sparse data conditions. DECKT employs a momentum-updated dual-encoder architecture where the primary encoder processes current input data while the momentum encoder maintains stable historical representations through exponential moving average updates. These encoders naturally form contrastive pairs through temporal evolution, effectively enhancing representation capabilities for low-frequency knowledge concepts without requiring destructive data augmentation operations that may compromise knowledge structure integrity. To preserve semantic consistency in learned representations, DECKT incorporates a graph structure constraint loss that leverages concept–question relationships to maintain appropriate similarities between related concepts in the embedding space. Furthermore, an adversarial training mechanism applies perturbations to embedding vectors, enhancing model robustness and generalization. Extensive experiments on benchmark datasets demonstrate that DECKT significantly outperforms existing state-of-the-art methods, validating the effectiveness of the proposed approach in alleviating representation challenges in sparse educational data.

## 1. Introduction

With the rapid advancement of online education platforms and intelligent learning systems, personalized learning has emerged as a central objective of modern education [1,2]. Personalization techniques have demonstrated effectiveness across diverse domains, from educational content delivery to user preference modeling in various applications [3,4]. Driven by breakthroughs in artificial intelligence and machine learning, adaptive learning systems have revolutionized educational delivery by automatically adjusting content difficulty, pacing, and instructional strategies based on individual learner characteristics and real-time performance indicators. These systems leverage learning analytics, recommendation engines, and intelligent tutoring systems to provide tailored instructional content and real-time feedback. Modern online education platforms integrate AI-powered assessment tools and adaptive algorithms to monitor learning progress continuously and optimize educational pathways dynamically, creating unprecedented opportunities for data-driven educational interventions [5,6].

Within this paradigm, knowledge tracing (KT) plays a pivotal role by modeling and predicting students’ evolving knowledge states over time, thereby enabling accurate learning assessment and informed instructional decision-making [7,8]. High-precision KT models are essential for capturing individual learning trajectories, understanding the complex interplay between knowledge components, and providing reliable cognitive state representations that drive personalized educational recommendations. Traditional knowledge tracing techniques, such as Bayesian knowledge tracing (BKT) [9], rely on hidden Markov models to estimate student knowledge states through probabilistic inference, treating knowledge mastery as binary states transitioning between learned and unlearned conditions. While effective for simple learning scenarios, these classical approaches struggle to capture the complexity of real-world learning behaviors, particularly the non-linear dynamics of knowledge acquisition, forgetting phenomena, and prerequisite relationships between knowledge components. Recent advances in deep learning have revolutionized this field, leading to sophisticated models based on recurrent neural networks, attention mechanisms, and memory-augmented structures [10,11]. These neural approaches demonstrate significant improvements in modeling capability and predictive accuracy by leveraging large-scale learning interaction data and capturing temporal dependencies in student learning processes. Deep KT models such as deep knowledge tracing (DKT) [12], dynamic key–value memory networks (DKVMNs) [11], and self-attentive knowledge tracing (SAKT) [10] have shown superior performance across diverse educational domains.

However, when confronted with data sparsity challenges, these methods still exhibit limited effectiveness in modeling low-frequency questions and knowledge concepts, particularly in realistic educational settings where student interaction patterns follow long-tail distributions. The fundamental issues stem from two critical aspects that significantly impact model performance in sparse data environments. On one hand, existing methods predominantly rely on predefined knowledge concept labels, treating all questions under the same concept as equivalent representations and overlooking fine-grained variations in content complexity, difficulty levels, and semantic expressions [9,11,12]. On the other hand, current approaches lack effective mechanisms to leverage structural relationships and contextual information within knowledge graphs, missing opportunities to exploit the hierarchical nature of knowledge domains and the semantic connections between related concepts. Without proper structural support and cross-concept semantic enrichment, low-frequency knowledge concepts suffer from insufficient training signals, leading to poor representation quality, increased prediction uncertainty, and degraded model generalization in sparse data environments.

To tackle these challenges, graph neural networks (GNNs) have been introduced to capture concept relations through structured graphs [13,14,15,16]. GNN-based approaches represent knowledge concepts and questions as nodes in a graph, with edges encoding relationships such as prerequisite dependencies and semantic similarities. This graph-based modeling enables the capture of high-order interactions and structural dependencies that are not apparent in traditional sequential models. While GNN-based methods enhance node representations by modeling semantic dependencies and propagating information through graph structures, they often suffer from limited message propagation on low-degree nodes, leading to degraded embeddings for low-frequency knowledge concepts. The message passing mechanism tends to favor well-connected nodes that receive rich information from multiple neighbors, while sparsely connected nodes struggle to obtain sufficient contextual information for effective representation learning. Their heavy reliance on explicit labels and predefined graph structures further constrains representation improvement in weakly supervised scenarios, and many GNN-based KT models assume static structures that fail to adapt to the dynamic nature of knowledge acquisition.

Recently, contrastive learning has shown significant promise in alleviating data sparsity challenges across various domains [17,18,19], demonstrating particular effectiveness in scenarios where labeled data are limited and representation quality is crucial. By constructing positive and negative sample pairs and optimizing embeddings to maximize agreement between similar instances while minimizing agreement between dissimilar ones, contrastive learning can extract rich semantic information from unlabeled data and improve model robustness. In knowledge tracing, existing contrastive approaches primarily focus on two strategies: graph structure-based contrasting through multi-view representations [20,21], and heterogeneous graph modeling with auxiliary tasks that incorporate additional supervision signals [22,23,24]. The first category typically employs techniques such as node masking, edge dropping, and subgraph sampling to create augmented views, then applies contrastive losses to align representations across these views. The second category focuses on incorporating auxiliary information such as textual descriptions and difficulty levels to create richer representations through multi-task learning frameworks. However, these methods rely heavily on explicit data augmentation operations that may destroy critical knowledge structures in sparse educational data, potentially severing knowledge concept dependency paths and disrupting hierarchical relationships. These issues collectively exacerbate representation quality imbalance, significantly constraining model performance for low-frequency knowledge concepts.

To address the above challenges, this paper proposes Dual-Encoder Contrastive Knowledge Tracing (DECKT). Inspired by momentum-based contrastive learning frameworks such as MoCo [25], DECKT constructs a dual-branch architecture comprising a primary encoder using graph attention networks (GATs) for modeling question–concept interactions, and a momentum encoder updated through the exponential moving average (EMA) that maintains a stable representation space. The primary encoder actively learns from training data, while the momentum encoder provides consistent representations that serve as reliable contrastive targets. This design generates differentiated contrastive views without explicit data augmentation, preserving knowledge structure integrity while improving representation quality imbalance for low-frequency knowledge concepts. DECKT incorporates a momentum-based dual-encoder architecture that avoids destructive data augmentation while providing robust contrastive learning through the natural evolution of model parameters. Additionally, a graph structure constraint loss function maintains semantic consistency between questions and knowledge concepts in the embedding space, enforcing appropriate similarities between related concepts while ensuring sufficient separation between distinct concepts. Furthermore, an adversarial training mechanism enhances model robustness through min–max optimization, introducing carefully calibrated perturbations to input representations that improve generalization in sparse data environments. Together, these components create a comprehensive framework that addresses both representation learning challenges and structural modeling requirements for effective knowledge tracing.

The main contributions of this paper are as follows:We propose DECKT, a novel dual-encoder contrastive learning approach that improves representation quality imbalance through momentum-based contrastive view generation, avoiding knowledge structure destruction while enhancing low-frequency knowledge concept representations.We design a graph structure constraint loss function to maintain semantic consistency between questions and knowledge concepts, and introduce an adversarial training mechanism to improve model robustness in sparse data environments.Extensive experiments on multiple large-scale datasets demonstrate that DECKT achieves superior and stable performance compared to existing methods in addressing representation challenges and modeling low-frequency knowledge concepts.

## 2. Related Work

### 2.1. Knowledge Tracing

Knowledge tracing serves as a fundamental technique in intelligent educational systems, aiming to model the evolution of learners’ knowledge states to predict their future learning performance. The field has undergone a significant transformation from traditional statistical approaches to deep learning methodologies. Early Bayesian knowledge tracing (BKT) [9] employed hidden Markov models to characterize binary state transitions of knowledge mastery, yet its simplified assumptions constrained the ability to capture complex learning processes. The emergence of deep learning techniques introduced revolutionary breakthroughs to this domain, with deep knowledge tracing (DKT) [12] incorporating LSTM networks to handle variable-length sequences and capture long-term dependencies. To address the knowledge concept representation confusion inherent in DKT, dynamic key–value memory networks (DKVMNs) [11] leverage external memory mechanisms to separately store knowledge concept representations and learning states. The successful integration of attention mechanisms further advanced the field, with self-attentive knowledge tracing (SAKT) [10] utilizing transformer architectures for parallel sequence processing, and context-aware attentive knowledge tracing (AKT) [26] incorporating forgetting factors to improve long sequence modeling capabilities. Recent advances include alternate autoregressive knowledge tracing (AAKT) [27] and cold-start knowledge tracing with kernel bias and cone attention (csKT) [28] to overcome limitations in sequence modeling.

The development of graph neural network techniques has motivated researchers to focus on modeling structured relationships among knowledge concepts. Graph-based knowledge tracing (GKT) [14] pioneered the integration of graph convolutional networks to model dependencies between knowledge concepts, establishing a research paradigm that combines graph structures with sequential modeling. Subsequent research has enriched the application of graph neural networks in knowledge tracing, including relation-aware self-attention for knowledge tracing (RKT) [29] which incorporates multiple relation types into self-attention mechanisms, and hierarchical graph knowledge tracing (HGKT) [30] which models complex knowledge dependencies through hierarchical exercise graphs. Recent developments encompass session graph-based knowledge tracing (SGKT) [31] for student performance prediction and joint graph convolutional network-based knowledge tracing (JKT) [32]. To address data sparsity challenges, GraphCA [33] improves knowledge tracing performance through counterfactual augmented learning, while dynamic graph embedding via meta-learning [34] provides novel insights for handling dynamic educational environments. Additionally, learning states-enhanced knowledge tracing [35] further improves model performance by simulating diversity in real-world learning processes. Modern knowledge tracing also confronts challenges in handling large-scale skill sets, heterogeneous learning patterns, and complex inter-skill associations, with dual-graph convolutional networks and positive/negative feature enhancement networks [36] addressing these challenges through simultaneous processing of dual-graph structures for knowledge concepts and student learning patterns.

Parallel to these developments, recent advances in large language models (LLMs) have opened new avenues for knowledge tracing applications. Initial efforts leverage LLMs for difficulty modeling through DCL4KT + LLM [37], which demonstrates how language models can enhance traditional contrastive learning frameworks by providing more accurate difficulty predictions. Researchers have also addressed cold-start problems via semantic feature extraction [38], enabling knowledge tracing systems to handle previously unseen questions more effectively. Further advances include structure-aware approaches like SINKT [39] that construct concept–question graphs using LLM-derived semantic relationships, representing a significant step toward inductive knowledge tracing. Domain-specific applications have also emerged, such as CEKT [40] for programming knowledge tracing, which harnesses code comprehension capabilities to improve assessment accuracy. While these approaches demonstrate the transformative potential of semantic understanding in educational contexts, they primarily address content-level enhancements rather than fundamental representation learning challenges. Nevertheless, existing approaches commonly encounter the sparsity challenge in knowledge concept–problem associations, particularly exhibiting deficiencies in representation learning for low-frequency knowledge concepts, thereby limiting predictive accuracy in practical applications.

### 2.2. Contrastive Learning

Contrastive learning, as a pivotal branch of self-supervised learning, learns effective data representations by maximizing attraction between similar samples while minimizing attraction between dissimilar samples, achieving remarkable success in computer vision and natural language processing domains. The Simple Contrastive Learning (SimCLR) framework [41] learns visual representations through data augmentation and contrastive loss functions, while Momentum Contrast (MoCo) [25] introduces momentum encoders and queue mechanisms to enhance training efficiency. As an important extension, Deep Graph Infomax (DGI) [42] learns node representations by contrasting graph patches with graph summaries, and Graph Contrastive Learning with Augmentations (GRACE) [43] maximizes mutual information between identical nodes across two augmented views.

In the knowledge tracing domain, contrastive learning applications demonstrate rapid development trends. Bi-graph contrastive learning-based knowledge tracing (Bi-CLKT) [20] employs a dual-layer contrastive learning framework through “exercise-to-exercise” and “concept-to-concept” structures, acquiring discriminative representations at both the node and graph levels, effectively addressing the limitation of traditional graph neural network methods that overemphasize node details while neglecting high-level semantic information. Difficulty-focused contrastive learning for knowledge tracing [37] integrates large language models for difficulty prediction, designing difficulty-centered contrastive learning methods that significantly enhance knowledge tracing model performance. Self-paced contrastive learning for knowledge tracing [44] achieves balance between global and local information through self-paced learning data augmentation and self-supervised contrastive learning, effectively mitigating model bias in sparse data environments. Weighted heterogeneous graph-based three-view contrastive learning [23] enhances tracking accuracy and robustness by integrating multi-view information of students and knowledge points. Contrastive learning for knowledge tracing (CL4KT) [21] proposes a contrastive learning framework that handles complexity in knowledge acquisition processes by revealing semantically similar or dissimilar samples in learning histories. Stable knowledge tracing with diagnostic transformer [24] constructs problem-level to knowledge-level architectures and proposes contrastive learning-based training algorithms focused on maintaining stability in knowledge state diagnosis. Knowledge tracing with contrastive learning and attention-based LSTM network (CALSKT) [45] constructs attention mechanism-based LSTM networks and extracts self-supervised signals from learners’ original learning histories through contrastive learning frameworks. However, existing work predominantly adopts contrastive learning frameworks from other domains without sufficient consideration of educational data characteristics, particularly regarding how to appropriately define positive and negative samples in knowledge tracing tasks to address knowledge concept–problem association sparsity challenges. Therefore, designing contrastive learning mechanisms suitable for knowledge tracing tasks, particularly solutions targeting knowledge concept–problem association sparsity problems, remains a research direction worthy of in-depth exploration.

## 3. Methods

### 3.1. Task Formulation

In knowledge tracing, the goal is to predict whether a student will correctly answer a future question based on their past interactions. Let $E=\{e_{1},e_{2},\ldots ,e_{n}\}$ denote the set of questions and $K=\{k_{1},k_{2},\ldots ,k_{m}\}$ the set of knowledge concepts. Each question $e_{i}$ may involve multiple concepts $k_{j}$, with their associations represented by an adjacency matrix $A\in R^{n\times m}$, where $A_{i,j}=1$ indicates that question $e_{i}$ is associated with concept $k_{j}$, and $A_{i,j}=0$ otherwise. A student’s interaction history is given as a sequence $S={\left\{\left(e_{t},r_{t}\right)\right\}}_{t=1}^{T}$, where $e_{t}$ is the question attempted at time *t*, and $r_{t}\in \left\{0,1\right\}$ indicates whether the response was correct.

To model question–concept dependencies, we construct a bipartite graph G=(V,E), where V=E∪K, and edges connect questions to their associated concepts as defined by *A*. The core task is to predict the probability that the student will answer the next question $e_{T+1}$ correctly, given their interaction history:(1)$$P(r_{T+1}=1|S,e_{T+1})$$

This requires learning effective embeddings of questions and concepts, and leveraging the student’s past performance to infer their mastery of future content.

### 3.2. Model Architecture

To alleviate representation quality imbalance caused by data sparsity, this paper proposes DECKT, a dual-encoder contrastive knowledge tracing model, as illustrated in Figure 1. DECKT employs a momentum-based dual-encoder architecture where the primary encoder, built upon graph attention networks (GATs), processes current input data and captures semantic interactions between questions and knowledge concepts, while the momentum encoder maintains stable representations through exponential moving average (EMA) updates. This design enables natural contrastive view construction through temporal parameter evolution, enhancing representation capabilities for low-frequency knowledge concepts without destructive data augmentation. The model optimizes a multi-objective loss function that integrates InfoNCE contrastive loss with a graph structure constraint loss, ensuring both contrastive informativeness and structural coherence in the learned embeddings. DECKT follows a two-stage training strategy: the representation learning stage jointly optimizes embeddings through contrastive and structural objectives, while the adversarial enhancement stage introduces calibrated perturbations to improve model robustness in sparse data environments. Upon training completion, the enriched question and concept representations are fed into a sequence prediction module to forecast student performance. The detailed implementation of each component is provided in the following sections.

### 3.3. Dual Encoder Module

The core of DECKT lies in its dual-encoder architecture that generates contrastive views through parameter evolution rather than explicit data augmentation. The architecture consists of two collaboratively trained encoders that naturally create diverse perspectives on the same graph structure. The primary encoder is built upon a multi-layer graph attention network, designed to capture complex high-order interactions between questions and knowledge concepts, as well as latent dependencies among concepts. The momentum encoder draws inspiration from the momentum update mechanism in contrastive learning [25], applying exponential moving average (EMA) updates to the parameters of the primary encoder to produce stable contrastive views, thereby reducing noise during representation learning. Unlike conventional contrastive methods that rely on explicit data augmentations, such as edge deletion or feature masking, to construct contrastive views, DECKT generates views directly from the intact graph structure. This design naturally preserves the original topology and improves representation stability under sparse data conditions.

#### 3.3.1. Primary Encoder

The primary encoder in DECKT is designed to learn robust node representations by capturing structural interactions between questions and knowledge concepts. DECKT adopts a multi-layer graph attention network (GAT) as its backbone, which enables the adaptive weighting of neighboring nodes through learnable attention mechanisms while preserving the structural flexibility of the knowledge graph.

Formally, given a question–concept adjacency matrix $A\in R^{n\times m}$, where *n* and *m* represent the number of questions and knowledge concepts, respectively, the primary encoder learns a mapping function $f_{\theta }$ that encodes structural information into node embeddings:(2)$$Z_{e_{i}}^{\theta },Z_{k_{j}}^{\theta }=f_{\theta }\left(A\right)$$
where $Z_{e_{i}}^{\theta }\in R^{n\times d}$ and $Z_{k_{j}}^{\theta }\in R^{m\times d}$ denote the *d*-dimensional embeddings of questions and knowledge concepts, respectively.

Within each GAT layer, the attention mechanism computes the normalized importance between a question node $e_{i}$ and its neighboring concept node $k_{j}$:(3)$$\alpha_{ij}=\frac{\exp\left(\mathrm{LeakyReLU}\left(a^{⊤}\left[WZ_{e_{i}}^{\left(l\right)}\;∥\;WZ_{k_{j}}^{\left(l\right)}\right]\right)\right)}{\sum_{k\in N(i)}\exp\left(\mathrm{LeakyReLU}\left(a^{⊤}\left[WZ_{e_{i}}^{\left(l\right)}\;∥\;WZ_{k_{k}}^{\left(l\right)}\right]\right)\right)}$$
where $W\in R^{d\times d^{\prime }}$ is a trainable transformation matrix, $a\in R^{2d^{\prime }}$ is a learnable attention parameter vector, ∥ represents vector concatenation, and N(i) denotes the neighborhood set for node *i*. This attention mechanism enables the model to adaptively focus on the most relevant knowledge concepts for each question.

The updated node representation is computed by aggregating information from its neighbors:  (4)$$Z_{e_{i}}^{\left(l+1\right)}=\sum_{j\in N(i)}\alpha_{ij}WZ_{k_{j}}^{\left(l\right)}$$
where $Z_{e_{i}}^{\left(l\right)}$ represents the embedding of node $e_{i}$ at layer *l*.

To capture diverse semantic relationships across multiple representational subspaces, DECKT employs multi-head attention with *H* parallel attention mechanisms, and the outputs are concatenated and linearly transformed to produce the final layer representation. To ensure stable training and mitigate gradient vanishing in deep architectures, DECKT incorporates residual connections and layer normalization:(5)$$Z_{e_{i}}^{\left(l+1\right)}=\mathrm{LayerNorm}\left(Z_{e_{i}}^{\left(l+1\right)}+Z_{e_{i}}^{\left(l\right)}\right)$$

This design maintains well-conditioned embedding spaces and enables effective information propagation across multiple GAT layers, facilitating the learning of high-order semantic relationships between questions and knowledge concepts.

#### 3.3.2. Momentum Encoder

Knowledge tracing requires temporal consistency in node representations to effectively model students’ evolving knowledge states. However, frequent gradient updates in the primary encoder may introduce instability in the embedding space, particularly under data sparsity where representations of low-frequency knowledge concepts are susceptible to noisy fluctuations.

To mitigate this challenge, DECKT incorporates a momentum encoder that maintains stable representations through parameter smoothing. The momentum encoder shares the identical GAT architecture as the primary encoder but is excluded from direct gradient-based optimization. Instead, its parameters are updated via the exponential moving average (EMA) of the primary encoder’s parameters, providing a temporally consistent view of the embedding space that serves as reliable contrastive targets.

Formally, let $\theta_{t}$ and $\phi_{t}$ denote the parameters of the primary and momentum encoders at training step *t*, respectively. The momentum encoder is initialized with the same parameters as the primary encoder and updated iteratively:(6)$$\phi_{0}=\theta_{0},\;\phi_{t}=m\cdot \phi_{t-1}+\left(1-m\right)\cdot \theta_{t}$$
where m∈[0,1) is the momentum coefficient that controls the temporal smoothing strength. A higher value of *m* increases the retention of historical information, while a lower value allows for faster adaptation to recent parameter changes.

The recursive update rule can be expanded to show the contribution of historical parameters:(7)$$\phi_{t}=\left(1-m\right)\sum_{i=0}^{t}m^{i}\cdot \theta_{t-i}$$

This formulation reveals that $\phi_{t}$ represents a weighted average of all historical primary encoder parameters, with exponentially decreasing weights for older updates.

Given the momentum encoder parameters $\phi_{t}$, the momentum encoder generates stable node embeddings:(8)$${\tilde{Z}}_{e_{i}}^{\phi },{\tilde{Z}}_{k_{j}}^{\phi }=f_{\phi }\left(A\right)$$
where ${\tilde{Z}}_{e_{i}}^{\phi }$ and ${\tilde{Z}}_{k_{j}}^{\phi }$ denote the question and concept embeddings produced by the momentum encoder, respectively. These embeddings maintain temporal consistency and serve as stable reference points for contrastive learning, effectively reducing the impact of gradient noise and improving representation quality for low-frequency knowledge concepts without compromising the structural integrity of the knowledge graph.

### 3.4. Prediction Layer

After obtaining enhanced node representations from the dual encoders, DECKT employs a prediction layer to model student knowledge states and forecast future performance. The prediction layer constructs interaction embeddings by combining question representations with response information:(9)$$x_{t}=Z_{e_{t}}^{\theta }+r_{t}\cdot E_{r}$$
where $Z_{e_{t}}^{\theta }$ is the question embedding from the primary encoder and $E_{r}$ is a learnable response embedding.

An LSTM network processes the interaction sequence to capture temporal dynamics:(10)$$h_{t}=\mathrm{LSTM}\left(x_{t},h_{t-1}\right)$$

For predicting performance on question $e_{T+1}$, the model combines the current knowledge state with the target question representation:(11)$$P\left(r_{T+1}=1|S,e_{T+1}\right)=\sigma \left(W_{p}\left[h_{T}\;∥\;Z_{e_{T+1}}^{\theta }\right]+b_{p}\right)$$
where σ is the sigmoid function, and $W_{p}$ and $b_{p}$ are learnable parameters. The model is trained using binary cross-entropy loss to optimize prediction accuracy.

### 3.5. Representation Optimization Module

To alleviate representation quality imbalance caused by data sparsity, DECKT incorporates a joint optimization strategy combining contrastive learning and structural consistency constraints. This design enhances the discriminability of node embeddings while maintaining semantic alignment with the underlying knowledge graph structure.

Contrastive Learning Objective. DECKT adopts a contrastive learning objective based on the InfoNCE loss to enhance representation discriminability. The objective encourages embeddings of the same node from the primary and momentum encoders to be similar while distinguishing different nodes. For question nodes, the contrastive loss is computed as:(12)$$L_{\mathrm{question}}=\sum_{i\in B}-\log\frac{\exp(\mathrm{sim}\left(Z_{e_{i}}^{\theta },{\tilde{Z}}_{e_{i}}^{\phi }\right)/\tau )}{\sum_{j\in B}\exp\left(\mathrm{sim}\left(Z_{e_{i}}^{\theta },{\tilde{Z}}_{e_{j}}^{\phi }\right)/\tau \right)}$$
where B denotes the batch of sampled nodes, $Z_{e_{i}}^{\theta }$ and ${\tilde{Z}}_{e_{i}}^{\phi }$ are the embeddings of question $e_{i}$ from the primary and momentum encoders, respectively, sim(·,·) denotes cosine similarity, and τ is the temperature parameter.

Similarly, the contrastive loss for knowledge concept nodes $L_{\mathrm{concept}}$ follows the same formulation. The total contrastive loss combines both components:(13)$$L_{\mathrm{contrast}}=L_{\mathrm{question}}+L_{\mathrm{concept}}$$

Graph Structure Constraint. To preserve semantic relationships between questions and knowledge concepts, DECKT introduces a graph structure constraint loss that minimizes embedding distances between connected nodes. For the question–concept bipartite graph G=(V,E), the structural loss is defined as:(14)$$L_{\mathrm{struct}}=\sum_{(e_{i},k_{j})\in E}{∥Z_{e_{i}}^{\theta }-Z_{k_{j}}^{\theta }∥}_{2}^{2}$$
where $(e_{i},k_{j})\in E$ indicates a connection between question $e_{i}$ and knowledge concept $k_{j}$, and $Z_{e_{i}}^{\theta }$ and $Z_{k_{j}}^{\theta }$ are the corresponding embeddings from the primary encoder. This constraint guides semantically related nodes to maintain proximity in the embedding space, enhancing representation quality for low-frequency knowledge concepts through better structural coherence.

The final optimization objective combines both components:(15)$$L_{\mathrm{total}}=L_{\mathrm{contrast}}+\lambda L_{\mathrm{struct}}$$
where λ is a hyperparameter balancing contrastive learning and structural consistency.

### 3.6. Adversarial Enhancement Module

To enhance model robustness for low-frequency knowledge concepts under sparse data conditions, DECKT incorporates an adversarial training mechanism that applies calibrated perturbations to the embedding space. This mechanism forces the model to learn robust representations that maintain consistency under adversarial conditions, improving generalization in sparse educational scenarios.

The adversarial training follows a min–max optimization framework:(16)$$\min_{\theta }\max_{∥\delta ∥\leq \epsilon }L_{\mathrm{adv}}\left(\theta ,Z^{\theta }+\delta ,{\tilde{Z}}^{\phi }\right)$$
where θ represents the primary encoder parameters, $Z^{\theta }$ denotes the primary encoder embeddings, ${\tilde{Z}}^{\phi }$ represents the stable momentum encoder embeddings, and δ is the adversarial perturbation bounded by ∥δ∥≤ϵ.

DECKT employs the Fast Gradient Sign Method (FGSM) to efficiently generate perturbations:(17)$$\delta =\epsilon \cdot \mathrm{sign}(\nabla_{Z^{\theta }}L_{\mathrm{contrast}}\left(Z^{\theta },{\tilde{Z}}^{\phi }\right))$$

The perturbations are applied separately to question and concept embeddings:(18)$${\hat{Z}}_{e_{i}}=Z_{e_{i}}^{\theta }+\delta_{e},\;{\hat{Z}}_{k_{j}}=Z_{k_{j}}^{\theta }+\delta_{k}$$

The adversarial loss is defined as the contrastive loss between perturbed primary encoder embeddings and stable momentum encoder embeddings:(19)$$L_{\mathrm{adv}}=L_{\mathrm{contrast}}\left({\hat{Z}}_{e_{i}},{\tilde{Z}}_{e_{i}}^{\phi }\right)+L_{\mathrm{contrast}}\left({\hat{Z}}_{k_{j}},{\tilde{Z}}_{k_{j}}^{\phi }\right)$$

This adversarial mechanism encourages robust representations that are particularly beneficial for low-frequency knowledge concepts where limited training data make the model susceptible to overfitting, thereby achieving enhanced generalization capability in sparse educational data environments.

### 3.7. Multi-Stage Training Strategy

To effectively leverage the complementary advantages of contrastive learning and adversarial training, DECKT employs a two-stage training strategy that progressively enhances representation quality and model robustness.

Stage 1: Representation Learning. The first stage focuses on learning high-quality node embeddings through the joint optimization of prediction accuracy, contrastive learning, and structural consistency:(20)$$L_{\mathrm{stage}1}=L_{\mathrm{pred}}+\lambda_{c}L_{\mathrm{contrast}}+\lambda_{s}L_{\mathrm{struct}}$$
where $L_{\mathrm{pred}}$ is the prediction loss (binary cross-entropy), $L_{\mathrm{contrast}}$ and $L_{\mathrm{struct}}$ are defined in the previous section, and $\lambda_{c}$ and $\lambda_{s}$ are weighting coefficients. During this stage, the primary encoder parameters θ are updated through gradient descent, while the momentum encoder parameters ϕ are updated via exponential moving average to provide stable contrastive targets.

Stage 2: Adversarial Enhancement. After the representation learning stage converges, the model enters the adversarial training stage to improve robustness by introducing carefully calibrated perturbations to the embedding space:(21)$$L_{\mathrm{stage}2}=L_{\mathrm{pred}}+\lambda_{c}L_{\mathrm{contrast}}+\lambda_{s}L_{\mathrm{struct}}+\lambda_{a}L_{\mathrm{adv}}$$
where $L_{\mathrm{adv}}$ represents the adversarial loss that applies min–max optimization to enhance model generalization under sparse data conditions, and $\lambda_{a}$ controls the strength of adversarial regularization.

This multi-stage approach ensures stable optimization by first establishing robust representations through contrastive learning, then refining decision boundaries through adversarial training, ultimately improving the model’s ability to handle low-frequency knowledge concepts in sparse educational datasets.

Algorithm 1 presents the complete DECKT training procedure. The algorithm employs a two-stage strategy: first optimizing node embeddings through combined prediction, contrastive, and structural losses to establish high-quality representations, then applying adversarial training to enhance model robustness and generalization in sparse educational data.
**Algorithm 1** Dual-Encoder Contrastive Knowledge Tracing (DECKT)**Require:** Graph *G*, dataset D, hyperparameters $\lambda_{c},\lambda_{s},\lambda_{a},m$**Ensure:** Embeddings $Z_{e_{i}}^{\theta },Z_{k_{j}}^{\theta }$, parameters θ  1:**Initialize:** θ, ϕ←θ  2:**Stage 1: Representation Learning**  3:**while** not converged **do**  4:      **for** batch (S,y)∈D **do**  5:          $Z_{e_{i}}^{\theta },Z_{k_{j}}^{\theta }\leftarrow f_{\theta }\left(G\right)$  6:          ${\tilde{Z}}_{e_{i}}^{\phi },{\tilde{Z}}_{k_{j}}^{\phi }\leftarrow f_{\phi }\left(G\right)$  7:          Compute $L_{\mathrm{pred}}$ using LSTM  8:          Compute $L_{\mathrm{contrast}}$ and $L_{\mathrm{struct}}$  9:          $L=L_{\mathrm{pred}}+\lambda_{c}L_{\mathrm{contrast}}+\lambda_{s}L_{\mathrm{struct}}$10:          Update θ to minimize L11:          ϕ←m·ϕ+(1−m)·θ12:      **end for**13:**end while**14:**Stage 2: Adversarial Enhancement**15:**while** not converged **do**16:      **for** batch (S,y)∈D **do**17:          Compute embeddings and losses as in Stage 118:          Generate adversarial perturbations19:          Compute $L_{\mathrm{adv}}$ with perturbed embeddings20:          $L=L_{\mathrm{pred}}+\lambda_{c}L_{\mathrm{contrast}}+\lambda_{s}L_{\mathrm{struct}}+\lambda_{a}L_{\mathrm{adv}}$21:          Update θ to minimize L22:          ϕ←m·ϕ+(1−m)·θ23:      **end for**24:**end while**25:**return **$Z_{e_{i}}^{\theta },Z_{k_{j}}^{\theta },\theta$

### 3.8. Theoretical Analysis

In sparse educational graphs, low-frequency knowledge concepts often suffer from unstable representations due to limited interaction data. Traditional contrastive learning approaches commonly rely on perturbations of the graph structure (e.g., edge sampling or node masking) to construct augmented views. However, such perturbations may disrupt the semantic integrity of item–concept relationships, particularly degrading the modeling of infrequent nodes.

To address this challenge, we adopt a momentum-based contrastive learning framework, where a momentum encoder is updated via an exponential moving average (EMA) of the primary encoder’s parameters:$$\phi_{t}=m\cdot \phi_{t-1}+\left(1-m\right)\cdot \theta_{t}$$

This update mechanism serves as a temporal smoothing process that accumulates historical information and suppresses gradient noise, which is particularly beneficial for low-frequency nodes. The contrastive objective is optimized using InfoNCE loss between embeddings from the primary and momentum encoders, where the stable representations generated by the momentum encoder effectively mitigate the risk of noisy or collapsed positives in sparse regimes.

From an information-theoretic perspective, prior work has established that [25,46] under continuity and view-stationarity assumptions, the InfoNCE loss provides a lower bound to the mutual information $I(z,z^{+})$ between positive view pairs, thereby promoting discriminative representations through mutual information maximization. Furthermore, since the momentum encoder evolves slowly without receiving direct gradients, it provides a quasi-static target view during training, which aligns with the convergence assumptions in recent theoretical analyses of contrastive learning. Together, these properties endow the proposed method with both practical robustness and theoretical guarantees for representation learning under data sparsity.

## 4. Experimental Design

### 4.1. Datasets and Preprocessing

To comprehensively evaluate the model’s performance across different educational scenarios, this paper conducts experiments on four representative datasets, with basic information shown in Table 1.

**ASSISTments2009 (ASSIST09)** (https://sites.google.com/site/assistmentsdata/home/2009-2010-assistment-data), accessed on 26 May 2025: From the ASSISTments online tutoring platform during the 2009–2010 academic year, recording students’ interaction trajectories on mathematics problems; widely used as a standard benchmark in knowledge tracing research.**ASSISTments2012 (ASSIST12)** (https://sites.google.com/view/assistmentsdatamining/dataset), accessed on 26 May 2025: Sourced from the 2012–2013 academic year, with larger data volume and more complex concept distribution; suitable for evaluating model scalability and generalization capability.**Slepemapy.cz** (https://pslcdatashop.web.cmu.edu/DatasetInfo?datasetId=1891), accessed on 26 May 2025: From an adaptive geography learning platform focusing on spatial knowledge acquisition tasks; used to test the model’s adaptability in concept modeling for non-mathematical domains.**EdNet** (https://github.com/riiid/ednet), accessed on 26 May 2025: Released by the Korean Santa educational platform, containing approximately 27 million interaction records from about 130,000 students, one of the largest publicly available educational datasets currently; commonly used for large-scale knowledge tracing and personalized learning research.Statics2011 (https://pslcdatashop.web.cmu.edu/DatasetInfo?datasetId=507), accessed on 26 May 2025: From Carnegie Mellon University’s Engineering Statics course during Fall 2011, containing student interaction records from the Open Learning Initiative (OLI) platform. This dataset focuses on engineering mechanics concepts and represents a typical STEM education scenario with complex multi-step problems involving force analysis, moment calculations, and structural mechanics.

### 4.2. Baseline Methods

To comprehensively evaluate model performance, this paper selects representative methods as comparison baselines, with detailed information as follows:**DKT** [12]: Uses LSTM networks to encode students’ knowledge states, treats the learning process as a sequence modeling problem, and predicts student performance on future exercises by updating hidden states.**DKVMN** [11]: Dynamically models students’ mastery of different knowledge concepts using two distinguishable memory matrices, capturing the evolution of knowledge states through reading and writing mechanisms.**SAKT** [10]: Utilizes self-attention mechanisms to capture relationships between exercises and knowledge concepts, effectively capturing long-range dependencies in students’ historical exercise sequences, suitable for modeling long sequential interaction information.**Deep-IRT** [47]: Combines deep learning with Item Response Theory (IRT), introducing student ability and exercise difficulty parameters in the prediction process, enhancing model interpretability.**GIKT** [13]: Utilizes graph structures to represent relationships between student interaction data and knowledge concepts, modeling dependencies between knowledge concepts through graph neural networks.**DKT + Forgetting** [47]: Introduces forgetting curve theory based on DKT, simulating students’ behavior of forgetting knowledge over time, making prediction results more aligned with real learning patterns.**GKT** [14]: Directly operates on knowledge graphs using graph neural networks, explicitly incorporating prior knowledge structure information, thereby enhancing the ability to model students’ knowledge states.**KTM** [48]: Models knowledge tracing tasks as feature interaction problems, integrating various sparse features such as students, exercises, and knowledge concepts using factorization machine frameworks, featuring good generalization ability and sparse adaptability.**AKT** [26]: Based on multi-head self-attention mechanisms, introduces monotonic attention to model forgetting behavior, while incorporating prior relationship information between knowledge concepts to improve modeling accuracy.**HawkesKT** [49]: Focuses on fine-grained temporal span effects between interactions in the learning process.**CL4KT** [21]: Adopts a contrastive learning paradigm, enhancing knowledge representation quality through positive and negative sample contrast mechanisms, improving the model’s ability to recognize similar learning patterns.**CDKT** [50]: Integrates the contrastive learning framework into the DKT model, aiming to enhance the quality of student representations through contrastive learning mechanisms, thereby more accurately modeling students’ knowledge states and learning behaviors.

### 4.3. Experimental Setup

The experiments in this paper are implemented based on the PyTorch 1.9.0 framework, with all experiments conducted on an NVIDIA A100 GPU equipped with 40 GB of memory. To ensure experimental reproducibility, this paper uses a fixed random seed (42) to initialize all model parameters. The main model employs a two-layer graph neural network as the encoder, with 2 attention heads per layer, an output dimension of 32 for each attention head, resulting in a final 64-dimensional node embedding representation. During training, this paper uses the Adam optimizer with an initial learning rate of 0.001 and applies a cosine annealing strategy for learning rate adjustment, with a minimum learning rate set to $1\times 10^{-5}$.

For the key components of the model, the momentum encoder’s update coefficient is set to 0.8, a relatively large value that ensures stability in feature representation. The temperature parameter τ for contrastive learning is set to 0.5, which has been validated as an effective value capable of producing sufficiently discriminative contrastive signals. In the adversarial training module, the perturbation upper bound ε is set to 0.01, with adversarial samples generated through 3 iterative steps, each with a learning rate of 0.001. These parameter settings have been thoroughly tuned on the validation set, achieving a good balance between model performance and training stability.

Training uses mini-batch gradient descent with a batch size of 64, with training data randomly shuffled in each epoch to increase sample diversity. This paper employs an early stopping strategy to prevent overfitting, terminating training when the AUC on the validation set shows no improvement for 10 consecutive training epochs. All experiments are repeated 5 times with the average performance reported to ensure statistical reliability of the results.

### 4.4. Overall Performance Comparison

This paper compares DECKT with various baseline methods across five datasets, as illustrated in Table 2 and Table 3. The results demonstrate that DECKT consistently achieves superior performance across all evaluation metrics. DECKT attains AUC scores of 0.7901, 0.7817, 0.7845, 0.7740, and 0.7893 on the ASSIST09, ASSIST12, Slepemapy.cz, EdNet, and Statics2011 datasets, respectively, showing significant improvements over existing methods. Across all datasets, our model outperforms the best baseline by up to 1.68 percentage points in AUC, demonstrating consistent superiority across diverse educational contexts. The experimental results validate DECKT’s excellent robustness and generalization capability, with the model maintaining stable advantages across various scenarios, from highly sparse datasets to large-scale educational platforms, thoroughly demonstrating its broad adaptability across different educational contexts and data characteristics.

### 4.5. Ablation Study Analysis

To evaluate the independent contributions of each DECKT component, we conduct systematic ablation experiments by removing the dual encoder architecture, adversarial training mechanism, and graph structure constraint loss. The results show clear hierarchical importance rankings across all five benchmark datasets, as shown in Figure 2. In the complexity analysis, we use the following notation: *L* denotes the number of GAT layers, |E| denotes the number of edges in the graph, |V| denotes the number of nodes, *d* denotes the embedding dimension, *H* denotes the number of attention heads, and |θ| denotes the total number of model parameters.

The dual-encoder architecture provides the largest performance contribution, with removal resulting in an average decrease of 1.13% AUC and 1.00% ACC. The impact is particularly significant on sparse datasets, with Slepemapy.cz and Statics2011 showing 1.25% and 1.15% AUC decreases, respectively, validating the effectiveness of momentum contrastive learning for low-frequency knowledge concepts. The computational complexity is O(L×|E|×d×H+|θ|) time and O(2|θ|+|V|×d+|E|) space overhead, where parameter replication of the momentum encoder doubles memory requirements, and EMA update operations require traversing all parameters at each training step. However, since this only affects the training phase without increasing inference costs, such an overhead is acceptable in practical deployment.

Adversarial training demonstrates consistent cross-dataset contributions, with performance stably decreasing by 0.58–0.68% AUC (average 0.63% AUC, 0.57% ACC) upon removal. Using a perturbation magnitude of ϵ=0.05 achieves an optimal robustness–performance trade-off by simulating natural noise in student behavioral data, enhancing the model’s adaptability to real deployment environments. This component incurs O(L×|E|×d×H+|V|×d) time complexity and O(|V|×d) additional space overhead, where perturbation generation requires extra gradient computation and backpropagation, increasing the training time by approximately 60–80%. While the storage overhead for perturbation vectors is relatively small, it still requires careful management on large-scale graphs and can be further optimized through gradient accumulation and mixed-precision training.

The graph structure constraint shows the smallest absolute contribution (average 0.44% AUC, 0.40% ACC), yet remains crucial for maintaining semantic consistency, with more significant contributions in sparse knowledge graphs. This component operates efficiently with O(|E|×d) time complexity and negligible space overhead, as it only requires computing similarity between adjacent nodes without additional structural storage, making it the most computationally efficient among the three modules. Importantly, it is independent of specific curricula or languages, relying solely on underlying graph structure, thus demonstrating potential for cross-educational system generalization. For scalability challenges on large-scale educational platforms, the current model controls overall complexity at the system design level by enforcing sparsity on concept–item adjacency matrices and limiting aggregation operations to local subgraph ranges. To further adapt to practical deployment requirements, future work can introduce more efficient encoders (such as graph convolutional networks replacing GAT), structure-aware subgraph sampling strategies, or leverage knowledge distillation for model compression and inference acceleration [51]. The graph structure constraint’s minimal computational footprint makes it particularly suitable for resource-constrained deployment scenarios, while its curriculum-agnostic nature facilitates seamless integration across diverse educational domains.

### 4.6. Embedding Visualization Analysis

To further evaluate the quality of representations learned by our model, we employed t-SNE dimensionality reduction techniques to visualize question node embeddings in two dimensions, as illustrated in Figure 3. The figure displays embedding distributions before (left) and after (right) applying contrastive and adversarial learning mechanisms, focusing on the 10 most sample-rich knowledge concepts (IDs 6, 28, 29, 31, 33, 47, 69, 70, 86, and 87) in the ASSIST09 dataset. The visualization results reveal that before training, question embeddings of the same knowledge concept (same color) demonstrate preliminary clustering tendencies, yet the overall distribution remains dispersed with irregular cluster shapes and ambiguous inter-class boundaries. Some knowledge concepts exhibit strong proximity relationships, such as ID 86 (yellow-green) with ID 6 (blue), and ID 69 (pink) with ID 31 (red) appearing significantly close in the embedding space; meanwhile, embeddings for ID 6 are particularly scattered, indicating poor representation quality and reflecting the model’s lack of distinct inter-class discriminative ability prior to training. After introducing contrastive learning and adversarial training, the structure of the embedding space improves markedly. The boundaries between different knowledge concepts become clearer, with more compact intra-cluster structures and significantly enlarged inter-cluster distances. For example, the previously mixed ID 86 and ID 6 achieve clear separation, while ID 69 and ID 31, though still adjacent, both demonstrate substantially enhanced internal cohesion. Additionally, previously dispersed knowledge concepts (such as ID 6 and ID 70) exhibit stronger intra-class cohesion. This indicates that through contrastive loss and perturbation mechanisms, the model significantly enhances intra-class similarity modeling and inter-class discriminability, improving the semantic structural consistency of embedding representations.

### 4.7. Prediction Visualization Analysis

Figure 4 presents a heatmap visualization of DECKT’s prediction performance on a representative student interaction sequence, demonstrating strong predictive calibration across 27 questions spanning six knowledge concepts (c37, c45, c54, c69, c70, c86). The visualization shows excellent high-confidence prediction accuracy, with all predictions ≥ 0.7 probability (11 instances including 0.87, 0.81, 0.84, and 0.86) correctly corresponding to positive student responses. Conversely, low-confidence predictions ≤ 0.4 (seven instances such as 0.21, 0.23, and 0.34) predominantly align with incorrect responses at an 85.7% rate, with only one boundary case (0.34 probability with correct response in concept c69) deviating from expected patterns. The model achieves an overall prediction accuracy of 81.5% (22/27), validating DECKT’s effectiveness in capturing student knowledge states. However, the visualization also reveals concept-specific performance variations: while concepts c37 and c70 demonstrate perfect prediction–response alignment, concept c45 shows systematic prediction errors (0.66–0.67 probabilities with incorrect responses), and concept c86 exhibits challenging low-mastery patterns with predominantly low probabilities and poor performance outcomes. These variations highlight DECKT’s ability to differentiate between well-mastered and problematic knowledge areas, providing educators with nuanced insights into student learning patterns across diverse conceptual domains while maintaining robust predictive reliability in high-confidence scenarios.

### 4.8. Hyperparameter Sensitivity Analysis

We conducted systematic experiments on the ASSISTments2009 dataset to assess the impact of key hyperparameters, momentum coefficient *m*, temperature parameter τ, adversarial perturbation strength ϵ, and contrastive loss weight $\lambda_{c}$, with the results illustrated in Figure 5. The momentum coefficient *m* shows optimal performance at 0.8 (AUC = 0.7901), achieving the best balance between stability and adaptability, while values of 0.7 or 0.9 result in slightly reduced performance. The temperature parameter τ performs optimally at 0.5 (AUC = 0.7901), as smaller values (τ=0.3) produce overly steep gradients and larger values (τ=0.7) yield insufficient discrimination. The adversarial perturbation strength ϵ exhibits asymmetric sensitivity with optimal performance at 0.05 (AUC = 0.7910), where higher values (ϵ=0.1) significantly degrade performance and lower values (ϵ=0.01) cause milder reduction. The contrastive loss weight $\lambda_{c}$ peaks at 1.0 (AUC = 0.7910) and remains stable up to 2.0, with excessive values interfering with knowledge tracing accuracy. The optimal configuration (m=0.8, τ=0.5, ϵ=0.05, $\lambda_{c}=1.0$) demonstrates consistent performance across datasets, validating the robustness of these hyperparameter settings for diverse educational scenarios.

### 4.9. Complexity Analysis

This section analyzes the time and space complexity of DECKT. Let |E| denote the number of questions, |K| the number of knowledge concepts, |V|=|E|+|K| the total nodes, |E| the number of edges, *d* the embedding dimension, *N* the number of training samples, *B* the batch size, and *T* the total number of training epochs. In the graph attention module, each layer performs edge-based attention computation and aggregation with complexity O(|E|·d), while the structural constraint loss involves distance calculations for all connected node pairs with the same complexity O(|E|·d). The contrastive loss employs batch-wise view construction with complexity $O(B^{2}\cdot d)$, and adversarial training includes perturbation generation and backpropagation with complexity O(B·d). Therefore, the time complexity per epoch is $O(|E|\cdot d+B^{2}\cdot d+B\cdot d)$, and the total training complexity is $O(T\cdot (|E|+B^{2})\cdot d)$. For space complexity, the model parameters from the encoder layers require $O(L\cdot d^{2})$ space, where *L* is the number of layers, the node embeddings occupy O(|V|·d) space, the graph structure requires O(|V|+|E|) storage, and the attention matrices require O(|E|) space, bounded by $O(|V{|}^{2})$ in dense graphs. The total space complexity is $O(L\cdot d^{2}+|V|\cdot d+|E|+B)$, where model parameters and attention matrices are the primary memory bottlenecks that can be mitigated through sparse computation and gradient checkpointing techniques.

## 5. Conclusions

This paper investigates the challenge of representation quality imbalance in knowledge tracing under data sparsity, a critical issue that significantly impacts the effectiveness of adaptive learning systems in real-world educational scenarios. To address this challenge, we propose DECKT, a dual-encoder contrastive learning framework that systematically improves representation quality for low-frequency knowledge concepts through three key innovations. First, DECKT employs a momentum encoder to generate stable contrastive views without destructive data augmentation, preserving the structural integrity of knowledge graphs while enabling effective contrastive learning. Second, the framework incorporates graph structure constraints to maintain semantic consistency between questions and knowledge concepts, ensuring that related concepts remain appropriately clustered in the embedding space. Third, adversarial training is integrated to enhance model robustness against input perturbations and improve generalization capabilities in sparse data environments. Extensive experiments on multiple benchmark datasets demonstrate that DECKT consistently outperforms state-of-the-art baselines across various evaluation metrics, with particularly notable improvements of up to 1.68 percentage points in AUC for scenarios involving low-frequency knowledge concepts. The ablation studies confirm the effectiveness of each component, while hyperparameter analysis validates the robustness of the proposed approach across different educational datasets. Future work may explore lightweight contrastive learning strategies to reduce computational overhead, advanced structural regularization techniques for better handling of extremely sparse concepts, and multi-modal extensions to incorporate additional educational features such as learning context and student profiles, thereby further enhancing model effectiveness and practical applicability in diverse educational applications.

## Figures and Tables

**Figure 1 entropy-27-00685-f001:**
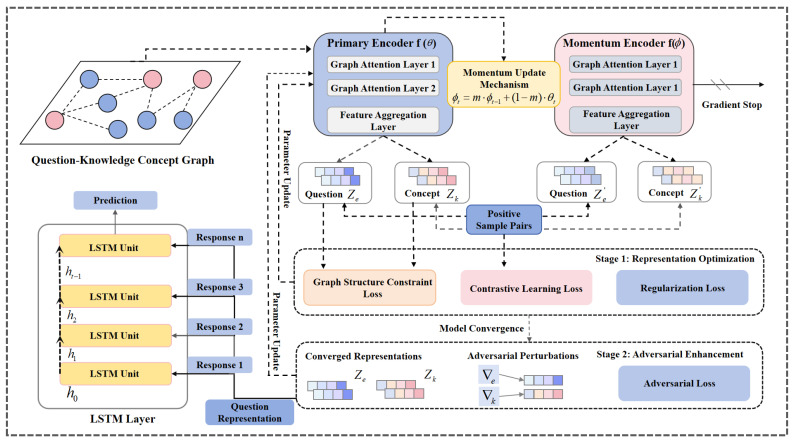
Overview of DECKT model framework. Blue indicates primary encoder, pink shows momentum encoder, yellow represents LSTM and momentum updates. Dashed arrows indicate momentum updates and interactions.

**Figure 2 entropy-27-00685-f002:**
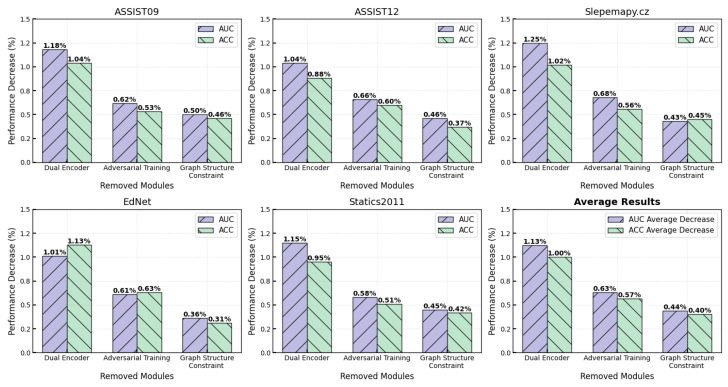
Ablation study analysis.

**Figure 3 entropy-27-00685-f003:**
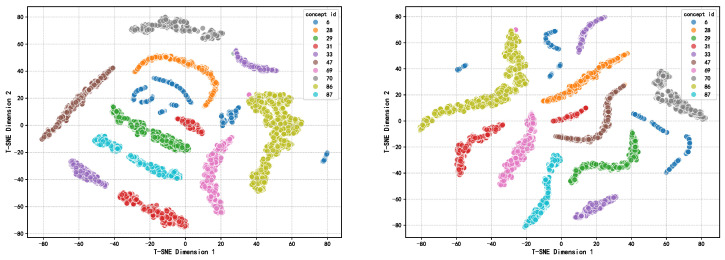
Embedding visualization analysis.

**Figure 4 entropy-27-00685-f004:**
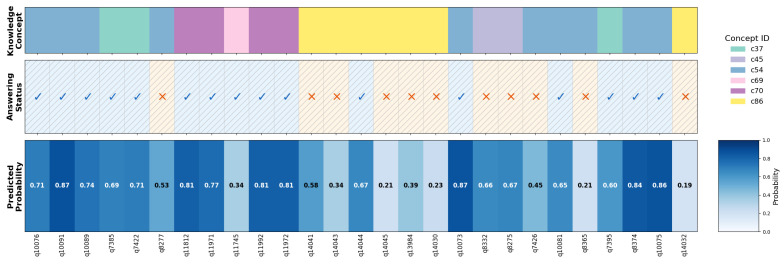
Prediction visualization analysis. The top row indicates the associated knowledge concepts for each question. The middle row shows the student’s actual answers, where ✓ indicates a correct response and × indicates an incorrect response. The bottom row presents the predicted probabilities of answering correctly, with darker blue indicating a higher predicted probability of a correct response.

**Figure 5 entropy-27-00685-f005:**
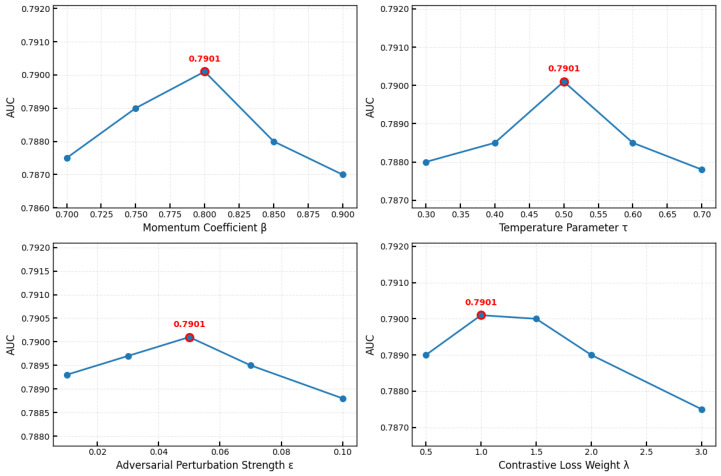
Hyperparameter sensitivity analysis. Red dots indicate the optimal parameter values that achieved maximum AUC.

**Table 1 entropy-27-00685-t001:** Dataset statistics.

Dataset	Students	Exercises	Knowledge Concepts	Interactions
ASSIST09	3.7 k	16.9 k	111	110.2 k
ASSIST12	25.3 k	50.9 k	245	879.5 k
Slepemapy.cz	81.7 k	2.9 k	1473	2877.5 k
EdNet	784.3 k	13.2 k	188	95,293.9 k
Statics2011	0.33 k	1.2 k	110	195.0 k

**Table 2 entropy-27-00685-t002:** AUC performance comparison across different models and datasets.

Model	ASSIST09	ASSIST12	Slepemapy.cz	EdNet	Statics2011
DKT	0.7525	0.7322	0.7512	0.7468	0.7698
DKVMN	0.7326	0.7057	0.7371	0.7303	0.7545
DKVMN-E	0.6742	0.6943	0.7237	0.7013	0.7197
SAKT	0.6894	0.6912	0.6739	0.6972	0.7123
Deep-IRT	0.7565	0.7365	0.7546	0.7461	0.7612
GIKT	0.7559	0.7523	0.7640	0.7245	0.7667
DKT + Forgetting	0.7573	0.7462	0.7574	0.7281	0.7623
GKT	0.7489	0.7345	0.7501	0.7344	0.7659
KTM	0.7353	0.7514	0.7421	0.7469	0.7639
AKT	0.7519	0.7649	0.7547	0.7423	0.7735
HawkesKT	0.7617	0.7669	0.7572	0.7546	0.7801
CL4KT	0.7616	0.7539	0.7465	0.7353	0.7683
CDKT	0.7733	0.7709	0.7545	0.7505	0.7745
DECKT	0.7901	0.7817	0.7845	0.7740	0.7893

**Table 3 entropy-27-00685-t003:** ACC performance comparison across different models and datasets.

Model	ASSIST09	ASSIST12	Slepemapy.cz	EdNet	Statics2011
DKT	0.7247	0.7371	0.7820	0.7184	0.7356
DKVMN	0.7189	0.7294	0.7859	0.7026	0.7298
DKVMN-E	0.6946	0.7255	0.7847	0.7032	0.7089
SAKT	0.6864	0.7216	0.7711	0.6748	0.6912
Deep-IRT	0.7268	0.7396	0.7823	0.7364	0.7489
GIKT	0.7371	0.7389	0.7366	0.7080	0.7401
DKT + Forgetting	0.7272	0.7373	0.7819	0.7023	0.7367
GKT	0.7178	0.7300	0.7799	0.7137	0.7303
KTM	0.7100	0.7412	0.7772	0.7159	0.7380
AKT	0.7199	0.7523	0.7827	0.7204	0.7438
HawkesKT	0.7305	0.7475	0.7823	0.7318	0.7515
CL4KT	0.7235	0.7215	0.7708	0.7181	0.7360
CDKT	0.7297	0.7019	0.7786	0.7239	0.7335
DECKT	0.7450	0.7753	0.8190	0.7490	0.7527

## Data Availability

Data are available on request due to restrictions, e.g., privacy or ethical. The data presented in this study are available on request from the corresponding author.

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
