# Peer review of "A Dual-Encoder Contrastive Learning Model for Knowledge Tracing"

_entropy, 2025, doi:10.3390/e27070685_

Round 1

Reviewer 1 Report

Comments and Suggestions for Authors

Reviewer Comments

  1. The authors have developed a comprehensive dual-encoder contrastive learning framework for knowledge tracing that addresses the critical challenge of data sparsity in educational systems. The momentum-based approach avoids destructive data augmentation while maintaining the benefits of contrastive learning, demonstrating practical applicability across four benchmark educational datasets with consistent improvements of up to 1.68 percentage points in AUC.
  2. The momentum encoder architecture represents a notable technical contribution to knowledge tracing. The exponential moving average update mechanism provides stable representations for low-frequency knowledge concepts without compromising knowledge graph structure integrity. This design naturally creates contrastive views through temporal parameter evolution, which is particularly innovative for educational data mining applications.
  3. The experimental evaluation is thorough, utilizing diverse educational datasets (ASSIST09, ASSIST12, Slepemapy.cz, EdNet) spanning different domains and scales. The authors demonstrate measurable improvements across multiple metrics, with particularly strong performance on sparse data scenarios. The comprehensive ablation studies, embedding visualizations, and hyperparameter sensitivity analysis provide convincing evidence of the method's effectiveness.
  4. While the paper presents a solid methodology, the theoretical foundation could be strengthened. The authors should provide more rigorous analysis of why momentum-based contrastive learning specifically benefits low-frequency knowledge concepts, particularly regarding the information-theoretic principles underlying representation learning in sparse educational environments and the mathematical guarantees of convergence in their dual-encoder framework.
  5. The integration of graph attention networks with structural constraint loss for maintaining semantic consistency is innovative. However, the authors should discuss potential scalability challenges when extending this approach to larger educational platforms with millions of students and questions, or different pedagogical contexts beyond the current Western and Asian educational systems evaluated.
  6. The comparison with state-of-the-art knowledge tracing methods provides good baseline validation. Nevertheless, the paper would benefit from ablation studies that isolate the individual contributions of each component: dual-encoder architecture, graph structure constraints, and adversarial training to the overall performance gains, as well as computational overhead analysis for each module.
  7. The practical implications showing consistent performance across datasets with varying sparsity levels demonstrate real-world applicability. The authors provide convincing evidence of their system's effectiveness in handling low-frequency knowledge concepts, which is a persistent challenge in personalized learning systems. The embedding visualization analysis particularly strengthens the paper's contribution to educational technology literature.
  8. Although the paper covers relevant prior work in knowledge tracing and contrastive learning, it lacks integration with recent advances in large language models for education and their applications in personalized learning systems. To better position this work within current research trends and enhance its theoretical grounding, the authors should cite and discuss the following recent developments:

    - UIFRS-HAN: User interests-aware food recommender system based on the heterogeneous attention network. Engineering Applications of Artificial Intelligence. 2024 Sep 1;135:108766.
    - A novel healthy food recommendation to user groups based on a deep social community detection approach. Neurocomputing. 2024 Apr 1;576:127326.

    - DistillHGNN: A Knowledge Distillation Approach for High-Speed Hypergraph Neural Networks. InThe Thirteenth International Conference on Learning Representations (2025).

Reviewer 2 Report

Comments and Suggestions for Authors

Knowledge tracing is a challenging but important topic worthy of further research. It is important as future performance can be predicted based on the knowledge states of the learners, while there are existing difficulties like data sparsity challenges that prevent it from being adopted.

In this manuscript, the authors presented a contrastive learning framework that improves knowledge state representation under sparse data conditions. Dual-Encoder Contrastive Knowledge Tracing is deployed to overcome the above challenges. The experimental results are promising, and support that the proposed system outperform the existing ones.

As the focus of this manuscript is on different educational scenarios, four representative datasets have been tested in the evaluation. A total of 12 other methods are compared against these four datasets.

Suggestion:

- The authors have proposed a novel approach to solve this interesting problem.

- The 12 other methods that are compared are very comprehensive, but, it would be desirable if more datasets (currently four) that cover even more different scenarios can be included in the comparison.

- It would also be very desirable if clear graphical presentation for Figure 5 can be used in the revised version.

- Currently, all the results are displayed visually in the Figures. It may be easier for the readers to do further analysis if tables can also be included in the experiement result section.
